# Exposure to Traumatic Events at Work, Post-Traumatic Symptoms, and Professional Quality of Life among Italian Midwives: A Cross-Sectional Study

**DOI:** 10.3390/healthcare12040415

**Published:** 2024-02-06

**Authors:** Alice Guzzon, Giulia Nones, Claudia Camedda, Yari Longobucco

**Affiliations:** 1Department of Medical and Surgical Sciences, University of Bologna, 40138 Bologna, Italy; giulia.nones@studio.unibo.it (G.N.); claudia.camedda2@unibo.it (C.C.); 2IRCCS Azienda Ospedaliero-Universitaria di Bologna, 40138 Bologna, Italy; 3Department of Health Sciences, University of Florence, 50139 Florence, Italy; yari.longobucco@unifi.it

**Keywords:** midwives, post-traumatic stress disorder, professional quality of life, compassion fatigue, compassion satisfaction

## Abstract

Background: The aim of this study is to investigate the potential occurrence of post-traumatic stress disorder (PTSD) symptoms, following exposure to traumatic events, in Italian midwives and their consequent influence on the quality of midwives’ professional lives. In addition, data were collected on the major traumatic events described by midwives. Method: A cross-sectional study related the socio-demographic characteristics of 286 midwives with the scores obtained on two assessment scales, one for post-traumatic stress disorder (IES-R) and the other for quality of life (ProQOL V). The percentage of midwives who obtained a score higher than the predetermined threshold value in both questionnaires was noted, and the correlations that emerged were highlighted. Through this qualitative method, their significant work-related traumatic events were investigated to finally detect the prevalence percentage of each category. Results: The proportion of midwives scoring higher than 33 on the IES-R scale, indicating a higher likelihood of PTSD, was 48.6%. Freelancers or outpatient clinic midwives had lower mean IES scores (*p* = 0.049). A significant inverse correlation was observed between age and IES-R score and between the compassion satisfaction subscale and time since completing education (*p* = 0.028). A comparison between the IES-R and ProQOL scales showed a statistically significant correlation (*p* < 0.001), in particular, between the burnout (BO) (*p* < 0.001) and secondary traumatic stress (STS) (*p* < 0.001) subscales. The thematic categorization of traumatic events included mother/child death, mother/child medical complications, relational problems with patients or team members, and organizational problems/medical staff’s inexperience. Conclusions: The emerging data may confirm the data in the literature, namely those showing that midwives are prone to developing work-related PTSD, particularly due to their exposure to traumatic events such as maternal and neonatal death.

## 1. Introduction

Trauma is an emotional response to a shocking and overwhelming event or experience that occurs suddenly, usually involving a serious threat to the physical, emotional, or psychological well-being and safety of individual victims and their loved ones [1].

According to the American Psychological Association, in the Diagnostic and Statistical Manual of Mental Disorders (DSM-V), post-traumatic stress disorder (PTSD) is characterized by typical symptoms resulting from exposure to a traumatic factor—an event perceived as a threat to oneself or someone else’s life. To diagnose PTSD, clinical interviews and psychodiagnostic instruments are essential, as no single instrument alone is sufficient for diagnosis without its analysis in the context of a clinical interview [1,2].

While PTSD has been extensively studied in nurses, there has been limited exploration in midwives. However, the literature suggests that midwives, particularly during labor and delivery care, may be exposed to traumatic events leading to PTSD symptoms [3,4,5,6]. In the midwifery field, a traumatic birth involves serious injury, real or perceived, or maternal and/or neonatal death. Midwives may witness indirect exposure to the traumatic perinatal events experienced by women, which has been associated with negative psychological reactions, including PTSD [1,7].

Healthcare professionals experiencing PTSD symptoms may exhibit empathic impairment and emotionally distant caregiving, affecting their approach to patients and clinical decision-making abilities. Midwives, in particular, may feel personally upset and perceive negative impacts on both personal and professional aspects of their lives following traumatic perinatal events [5,6]. Therefore, PTSD symptoms could have a negative impact on their approach to the women they care for and influence their clinical decision-making abilities. Relationships with women under their care are identified as a potential vulnerability factor for midwives [5,6], as the connection established between the midwife and the woman giving birth is highly influential. Any complication or loss during childbirth could affect the professional and expose them to emotional distress, especially given that guilt and a sense of responsibility are felt more acutely among midwives than among other healthcare professionals due to the time spent with the woman during labor [8]. Factors influencing the onset of a traumatic stress response encompass the characteristics of the event, the organizational context, parenting issues, perceived coworker behaviors, perceived responsibility and blame, and personal relevance [6].

Symptoms of PTSD play a crucial role in the personal and professional lives of healthcare professionals, affecting the quality of their professional care [9,10]. This quality is measured through compassion satisfaction (CS) and compassion fatigue (CF), with CF encompassing burnout (BO) and secondary traumatic stress (STS) [11].

The concept of CS is characterized by positive feelings such as satisfaction in having positive feedback about one’s care and thus thinking that it is a pleasure to help others through one’s work. In contrast, CF considers the negative aspect of caregiving and includes two dimensions: burnout (BO) and secondary traumatic stress (STS) [11]. Burnout is a symptom of work-related stress and involves feelings, usually with a gradual onset, such as exhaustion, frustration, anger, and depression; it may reflect feelings of difficulty in coping with the work or performing it effectively, or it may be associated with a very high workload or an unfavorable work environment [11]. Secondary traumatic stress, on the other hand, refers to negative feelings driven by work-related fear and trauma. The development of problems due to exposure to the trauma of others can happen to people who have experienced traumatic and stressful events; in this case, it is called secondary exposure. Symptoms of STS are usually rapid in their onset and associated with a particular event, and they may include fear, difficulty sleeping, having images of the upsetting event come to mind, or avoiding things that remind one of the event [11].

Given these considerations, the aims of this paper are to examine the occurrence of PTSD symptoms in response to witnessing potentially traumatic events during the birth journey and to evaluate their consequent impact on the quality of midwives’ professional lives. In particular, we investigated the prevalence of PTSD symptoms; the level of compassion, satisfaction, and fatigue; and whether there were any correlations between these factors and the socio-demographic characteristics of the participants. Finally, we collected qualitative data to understand the types of traumatic phenomena that midwives experience.

## 2. Materials and Methods

This study was conducted in Italy from 30 June to 30 September 2022, with the approval of the Bioethics Committee of the University of Bologna (Prot. n. 83931 del 20 April 2022). The study design was a cross-sectional one, and sample estimation was performed through a comparison with a similar study, a 2017 Israeli study that enrolled 93 midwives [12]; 286 midwives were enrolled in the present study. Convenience enrollment took place through the dissemination of the link to the questionnaire on Google Forms^®^ forwarded via email to all provincial and interprovincial Orders of the Profession of Midwifery whose contacts were collected from the official National Federation of the Order of Midwifery Profession (FNPO) website and then through online groups of midwives, through e-mail addresses of midwifery coordinators in the various Italian provinces, and through empowering all respondents by asking them to personally advocate for the dissemination of the questionnaire among their fellow midwives. Anonymity was ensured because the questionnaire did not request the participant’s name. Informed consent to participate in the study was collected from all participants who completed the dedicated section within the questionnaire and was kept separately.

The questionnaire includes socio-demographic questions, two rating scales, and a section where participants describe traumatic events. These scales are the Impact of Event Scale-Revised (IES-R) [13] and the Professional Quality of Life (ProQOL V).

The IES-R, featured in the fifth edition of the DSM, comprises three subscales (avoidance, intrusion, and hyperexcitation) [4,5,6,14] and is widely used globally. It has been translated and validated in numerous languages, including Italian [2]. For this study, the Italian version was chosen, demonstrating good psychometric properties [15] and suitability for both research and clinical practice in Italy [16,17,18]. It comprises 22 items that capture the difficulties individuals may experience following stressful events. Participants rate their responses based on the perceived difficulty level over the past 7 days, using a 5-point scale ranging from 0 (not at all) to 4 (extremely). Summing the scores provides a single severity index ranging from 0 to 88. While the scale itself is not a diagnostic measure, a cutoff of 33 is commonly used to identify the potential presence of PTSD symptoms [4,5,6,19].

The ProQOL, developed by Stamm [11], is the most widely used scale for assessing the quality of professional life [11,12] and also in different Italian studies [20,21,22]. In this study, we utilized the Italian version of the V scale, developed by Gillian Jarvis and Fulvio Mazzacane. It comprises 30 statements, categorized into three subscales based on their correlation with compassion satisfaction, burnout, and compassion fatigue. Each item is rated using a 5-point scale ranging from 1 (rarely/never) to 5 (very often) based on how well the reported statement reflects the answer that was most true in the past 30 days. Scores are computed independently by summing the items within each subscale. The final score ranges from 0 to 50, with designated reference ranges indicating the impact on quality of life: low level (0–22), medium level (23–41), and high level (42–50) [11]. The detection of the presence of PTSD and STS, included in the ProQOL scale, gives a more precise indication of the nature of the trauma, i.e., whether it is work-related (STS) or not necessarily work-related (PTSD) [1,11].

Following a Shapiro–Wilk test study to determine whether the distribution was normal, a descriptive analysis was carried out. When appropriate, data were described using the absolute frequency, percentage, mean, and standard deviation. An inferential analysis was performed through an ANOVA test. After that, a linear regression test was adopted in order to verify the potential association between the variables. Firstly, data were studied with univariate models, and then all the variables that showed a significance level < 0.10 were included in a multivariate model.

The method used complies with ethical principles, and the paper and preparatory work were carried out using data analyzed in a totally anonymous manner.

## 3. Results

### 3.1. Sample Description

The characteristics of the sample are described in Table 1. This study included 286 midwives, the majority of whom fall within the age range of 20 to 40 years (69.8%), graduated more than 10 years ago (53.3%), and are employed in North Italy (73.7%). Most participants hold a bachelor’s degree in Midwifery (62.5%), adhere to the Christian faith (72.3%), work in labor suites (50.3%), and are either single (51.4%) or married (38.8%).

### 3.2. Analysis of Quantitative Data

From the analysis of responses to the IES-R scale, it emerged that 48.6% of the sample of midwives obtained a score higher than 33, which is associated with a higher probability of having PTSD. Comparing socio-demographic variables with scores on the IES-R scale through an ANOVA test revealed that midwives working as freelancers (mean 21.93 ± 19.0) or in outpatient clinics (mean 27.91 ± 18.40) had lower average values compared to colleagues working in other services (mean min 32.05, mean max 41) (*p* = 0.049) (Table 2). Furthermore, as respondents’ age increases, IES-R scores decrease (β = −2.508, *p* = 0.031). In fact, in the age group between 20 and 39 years, the average IES-R scores are higher than 35 points, while above 40 years, they are lower than 29 points (*p* = 0.043). There is no correlation with marital status, educational attainment, degree, religion, and geographic area of residence. From the comparison conducted through the total values obtained in the ProQOL scale and socio-demographic characteristics, no statistical correlation was found. However, when the ProQOL scale was divided into its three subscales—compassion satisfaction (CS), burnout (BO), and secondary traumatic stress (STS)—a statistically significant inverse correlation (β = −1.571, *p* = 0.028) emerged specifically with the completion of studies in the compassion satisfaction subscale using linear regression. This indicates that as the time elapsed since the completion of education increased, positive feelings regarding one’s work decreased. The results obtained in the ProQOL scale are reported in Table 3, which shows the percentage of midwives having a low (≤22), medium (23–41), or high (≥42) level of professional quality of life. In our findings, when divided into subscales, the participants showed a medium/high level of compassion satisfaction and a low/medium level of burnout and secondary traumatic stress. Finally, through linear regression, the two scales in the IES-R and ProQOL questionnaires were compared. The comparison of the total scores indicates that as the score in the IES-R scale increases, the total score in the ProQOL scale also increases (β = 0.283, *p* < 0.001). When the subscales are analyzed separately, it is seen that an increase in scores in the IES-R scale is associated with a particular increase in scores in the BO subscale (β = 0.084, *p* < 0.001) and the STS subscale (β = 0.212, *p* < 0.001) (Table 4).

### 3.3. Analysis of Qualitative Data

The final question in the questionnaire allows participants to describe the traumatic events they have encountered. These data were categorized thematically and subcategorized, and the percentage frequency was calculated. The thematic categories that emerged are as follows: mother/child death, mother/child medical complications, relational problems with patients or team members, and organizational problems/medical staff inexperience. The “other” category was used to encompass events not falling into the aforementioned categories (see Table 5). Among the participants, 38.1% reported having witnessed a mother or child’s death, with neonatal and fetal deaths outnumbering maternal deaths (86 cases vs. 21). Among the medical complications (37%), peripartum or in-labor complications (49 cases), such as postpartum hemorrhage, placenta abruption, and amniotic fluid embolism, were the most frequently reported. Episodes of mobbing and difficulties with other team members or patients had a traumatic impact on 9.76% of the participants, underscoring the negative influence of a poor work environment on midwives. Furthermore, as a subcategory within the “other” group, the SARS-CoV-2 pandemic emerged as a traumatic event for 2.1% of midwives.

## 4. Discussion

The results of this study showed a prevalence of PTSD of 48.6%. As for the repercussions on the quality of life, there were moderately low levels of compassion satisfaction and moderately high levels of burnout and secondary traumatic stress. For the midwifery profession, exposure to traumatic events related to care is quite common [8]. Some studies in the literature report percentages greater than 60% of professionals who have witnessed traumatic events. Specifically, 94% of midwives in Israel [12] and 67% in Australia [23] report having experienced a work-related traumatic event. In our sample, only 5 out of 286 midwives reported not having experienced a traumatic event, resulting in a percentage of 98.2% of witnesses to traumatic events. Of these, 48% also scored higher than 33 on the IES-R scale, meeting the criteria for the presence of a probable post-traumatic stress disorder (PTSD) following work-related events. Compared to the literature, the percentage of individuals testing positive for PTSD in our study is higher. Reported percentages in other studies range from 16% to 36% [3,4,12,23,24,25]. However, since the same scale for evaluating PTSD symptoms was not used in all studies, the discrepancy may be due to differences in the perception and measurement of post-traumatic stress disorder.

While the passage of years practicing a profession can increase one’s sense of competence and contribute to solidifying one’s personality [26], on the other hand, it can lead to greater exposure to potentially traumatic events [4]. In this study, as chronological age increases, IES-R test scores decrease; age seems to have a protective effect on professionals who over the years may have developed mechanisms to overcome trauma related to care. Over time, especially since completing their education, the scores on the compassion satisfaction subscale of the ProQOL decrease in midwives, highlighting how positive feelings about their work decrease with the passage of time since their last training.

The work setting is also correlated with a positive score on the IES-R test. Midwives working as freelancers or in outpatient clinics report lower average scores than colleagues working in emergency settings such as the delivery room (Table 2). However, the highest average scores were found in the group of participants who specified “other” as their work setting, indicating areas such as research, midwifery coordination, COVID vaccination hubs, universities, training, assisted reproductive technology, and day services. To understand this result, qualitative data from these participants were analyzed, revealing traumatic experiences previously lived in the clinical emergency setting, such as neonatal death or maternal/neonatal complications related to labor/delivery. So, it is reasonable to assume that the obtained score is due to the type of event experienced rather than the current work setting.

Being witnesses to traumatic events can have repercussions on professional quality of life. The literature suggests that individuals suffering from PTSD are likely to experience compassion fatigue, resulting in high levels of burnout and secondary traumatic stress [27]. This trend seems to be confirmed in the present study; as scores on the IES-R scale increase, scores on the ProQOL scale also increase, particularly in the subscales measuring compassion fatigue, i.e., burnout and secondary traumatic stress (Table 4).

As reported in the literature, in the description and classification of the main traumatic events experienced by midwives, the most recurring event is death. In fact, Cohen et al. (2017) [12] found that most events are related to emergencies (especially massive bleeding) and the death of the baby or mother. Similarly, in the study by Leinweber et al. (2017) [23], the most recurring traumatic events reported by midwives involve death, injuries, poor care, and lack of interpersonal respect. Other studies have described a variety of events during labor and delivery that can trigger traumatic stress responses [8,28,29]; these include not only obstetric emergencies but also “rough approaches” towards women by doctors and disrespectful interactions between healthcare personnel and women.

In this study, the recurring events reported by midwives involve maternal/neonatal death with multiple causes and medical complications suffered by the mother/baby. Although to a considerably lesser extent, relational problems with other team members and assisted women have also emerged, such as abuse of power by the attending doctor or aggression from the user.

## 5. Conclusions

The primary aim of this study is to investigate the prevalence of PTSD symptoms following exposure to work-related traumatic events and the consequent impact on the professional quality of life of Italian midwives.

The results align with the initial expectations, indicating a high prevalence of emotional distress, probable PTSD, and exposure to traumatic events related to care among Italian midwives. A significant 98.25% of the study sample reported witnessing a traumatic event in their professional context, emphasizing the need to recognize the exposure to work-related traumatic events of midwives as a professional stressor.

The process that midwives undergo after experiencing a severe obstetric event varies; for some, it involves a “recovery of professional self-image”, while for others, it leads to a change in professional identity and a search for roles beyond emergency obstetrics or the specialty itself.

Considering the widespread nature of the phenomenon, implementing support programs becomes necessary to mitigate the repercussions of traumatic birth events experienced by women and witnessed by midwives. Effective prevention strategies identified include organizational practices and individual support, encouraging interactions among colleagues, supervisors, and support institutions.

While providing essential psychological support for midwives facing traumatic births is crucial, field training should be promoted to reduce stress responses. Regular ProQOL assessments could be included to evaluate the need for compassion interventions and maintain high levels of compassion, vital for midwives.

Despite meeting initial expectations, the study has limitations. The use of only the IES-R questionnaire for PTSD diagnosis, which requires confirmation through clinical interviews, is one limitation. Additionally, the online questionnaire made it challenging to interpret responses categorized as “not evaluable”.

The sample size and composition limit the generalizability of the results. Representing only 1.65% of the total population of Italian midwives, with a higher representation from the North compared to the Center, South, and abroad, the study sample may not fully reflect the entire population.

In conclusion, despite the characteristics of the study sample and the instrument used, the emerging data confirm existing literature, indicating that midwives are at risk of developing work-related PTSD, particularly due to exposure to traumatic events such as maternal and neonatal deaths.

## Figures and Tables

**Table 1 healthcare-12-00415-t001:** Demographic and professional characteristics.

Group	N	Percentage %
**Age Range**		
20–29 years	108	37.7
30–39 years	92	32.1
40–50 years	34	11.6
>50 years	52	18.2
**Marital Status**		
Single	147	51.4
Married	114	38.8
Divorced	24	8.3
Widowed	1	0.3
**Provenance**		
North Italy	211	73.7
Center Italy	49	17.2
South Italy	19	6.7
Outside Italy	7	2.5
**Religion**		
Christian	206	72.3
Buddhist	4	1.4
Spiritual	1	0.3
Atheist	68	23.9
Agnostic	3	1
Non-believer	1	0.3
Other	2	0.7
**Education**		
Bachelor’s Degree enabling the Midwifery Profession (DPR 27/09/80)	33	11.5
University Diploma enabling the Midwifery Profession (DM2/04/2001)	33	11.5
Bachelor’s Degree in Midwifery (DM 509/99)	179	62.5
Master’s Degree in Obstetric Gynecological Sciences (DM 270/2004)	4	1.4
Master’s Degree in Nursing and Obstetric Gynecological Sciences	37	12.9
Time since completing education		
Newly graduated	65	22.8
5–10 years	68	23.9
>10 years	152	53.3
**Workplace**		
Delivery Room	144	50.3
Antenatal and Postnatal Ward	36	12.5
Obstetrics/Gynecology Outpatient Clinics	35	12.2
Private Practice	14	4.9
Delivery Room, Obstetric Ward, Outpatient Clinics	32	11.1
Other ^1^	25	8.7

^1^ Family Planning Clinic, Midwifery Coordination Laboratory Analysis, COVID-19 Vaccination, Research, University, Health Professions Facility, Oncological Prevention Center, Assisted Reproduction, Gynecology, Unemployed.

**Table 2 healthcare-12-00415-t002:** Mean scores obtained on the IES-R Scale and workplace.

Workplace	Mean Score on IES-R Test and SD	
Delivery Room	35.0 ± 20.0	
Antenatal and postnatal ward	32.0 ± 21.7	
Obstetrics/Gynecology Outpatient Clinics	27.9 ± 18.4	F [5;280] = 2.25, *p* = 0.049
Private Practice	21.9 ± 19.0	
Delivery Room, Obstetric Ward, Outpatient Clinics	32.1 ± 23.6	
Other	41.3 ± 26.3	

**Table 3 healthcare-12-00415-t003:** Professional Quality of Life Scale (ProQOL) assessment.

**Compassion Satisfaction**			
	**≤22** ^1^	**23–41**	**≥42**
**N**	8	184	94
**Percentage %**	2.8	64.2	32.9
**Compassion Fatigue: Burnout**			
	**≤22**	**23–41**	**≥42**
**N**	141	144	1
**Percentage %**	49.4	50.1	0.3
**Compassion Fatigue: Secondary Traumatic Stress**			
	**≤22**	**23–41**	**≥42**
**N**	189	95	2
**Percentage %**	66.3	3.9	0.7

^1^ Here, a score of **≤22** represents a low level; **23–41**, medium level; **≥42**, high level.

**Table 4 healthcare-12-00415-t004:** Linear regression between IES-R score and ProQOL score and its subscales.

	β	Standard Error	*p*-Value	95% Confidence Interval	
ProQOL Tot	38.560	0.773	<0.001	37.06895	40.111
ProQOL BO	20.664	0.616	<0.001	19.4517	21.8772
ProQOL STS	14.140	0.627	<0.001	12.90604	15.37193
ProQOL CS	−0.0165	0.019	0.395	−0.0547198	0.021676

**Table 5 healthcare-12-00415-t005:** Traumatic event classification.

Category/Subcategory	N	Percentage %
**Maternal/neonatal death**	**110**	**38.8**
Maternal death	21	-
Fetal/Neonatal death	85	-
Unspecified maternal/neonatal death	4	-
**Medical complications**	**108**	**37.7**
Maternal peripartum complications (labor, delivery, and immediate postpartum)	49	-
Intrapartum fetal complications	26	-
Events with neonatal complications	30	-
Undefined obstetric emergency	3	-
**Relationships with users and other team members**	**28**	**9.7**
Relationship with users	9	-
Relational issues with other team members	19	-
**Organizational issues/inexperienced medical staff**	**9**	**3.1**
Organizational issues	9	-
**Other**	**25**	**8.7**
Spontaneous/therapeutic abortion	7	-
COVID-19 pandemic	6	-
Legal issues	2	-
Undefined	10	-
**Absence of potentially traumatic events**	**5**	**1.7**
No traumatic events	5	-
**No response/non-interpretable**	**1**	**0.3**
Non-interpretable response	1	-
Total	286	

## Data Availability

The data presented in this study are available on request from the corresponding author (privacy reasons).

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
