# Peer review of "Exposure to Traumatic Events at Work, Post-Traumatic Symptoms, and Professional Quality of Life among Italian Midwives: A Cross-Sectional Study"

_healthcare, 2024, doi:10.3390/healthcare12040415_

Round 1

Reviewer 1 Report

Comments and Suggestions for Authors

Authors investigated the potential occurrence of posttraumatic stress disorder (PTSD) symptoms following exposure to traumatic events in Italian midwives and the consequent influence on the quality of professional life. As study result, midwives in Italy were prone to the risk of developing work-related PTSD, particularly due to exposure to traumatic events such as maternal and neonatal death.

The study is well designed and the manuscript is interesting.

My concern regards the language, especially the presentation of numbers. For example authors use commas rather than dots in Table 3 and in Table 1 when showing percentages. In English, in such cases, commas should be replaced by dots.

The same error can be found in results and abstract where commas occur where dots would be correct Please modify these errors and check other parts of the manuscript.

Author Response

Dear Reviewer,

Thank you for your suggestions. Following your guidance, we have corrected the errors with commas and periods in both tables and in the text.

Thank you again for your revision.

Kind Regards

Dr Guzzon Alice 

Reviewer 2 Report

Comments and Suggestions for Authors

The manuscript is focused on the problematic of trauma and posttraumatic symptoms. The manuscript has got a quantitative character, authors used adequate statistical methods and techniques for the data analyzing. The etxt is dividing into the chapters and subchapters typical for this kind of studies. I have got some comments, which could be helpful fort authors and they could improve the level of manuscript. The comments are presented below.

1. The kinds of information about sample size are insufficient, so please add concrete details about respondents, mainly from the table, where are presented demographic kinds of information.

2. The results are missing, probably the kinds of information presented in the tables belong into the findings and results. The better way is to add some words, which could be explaining the significance of the tables.

3. Please be uniform in the using of numbers after decimal point, the usual number is two.

I hope my comments are helpful.

Author Response

Dear Reviewer,

Thank you for your suggestions. Following your guidance, we have made some changes to our article.

We added more information about the sample between lines 150-154. We added more explanations regarding Table 3 between lines 174-178. We standardized the numbers after the decimal point in the tables. We chose to write one number after the point because some data have only one digit.

Thank you again for your helpful comments.

Kind regards,

Dr. Guzzon

Reviewer 3 Report

Comments and Suggestions for Authors

Dear Authors,

Congratulations on completing your study, which addresses an important topic. I would like to suggest a few things that could improve your submission.

Introduction
The introduction would benefit from a more explicit articulation of the research questions and hypotheses. This would help provide a clear framework for your study.

In section 2, particularly between lines 128-131, you have detailed the methods of analysis. To improve the clarity and coherence of your paper, it would be helpful to ensure that the results of these analyses are clearly presented in the Results section. In particular, could you include the tests of statistical significance that you mentioned? Including this information is critical for readers to understand the impact and validity of your findings. In addition, including the results of the linear regression you mentioned, along with a discussion of how the initial assumptions for using this method were met, would greatly strengthen your analysis. The p-value reported on line 162, while useful, does not provide a complete picture.

Regarding line 133, where you state that the data were collected anonymously, it would be helpful if you could elaborate on how this is consistent with the sampling procedure described on lines 93-100. At present, the connection between the anonymity of the data collection and the method of sending the emails is not immediately apparent.

Could you also provide a more detailed explanation of the scope and methods used to obtain informed consent?

For lines 143-145, it would be beneficial to include full information on the results of the ANOVA test.

On line 154, please clarify the approach used to assess statistical significance in this case.

The statement on lines 163-165 is intriguing; could you elaborate on its significance and implications?

Regarding Table 1, there appears to be some overlap in the age ranges listed. A revision of this table, both in terms of design for better readability and accuracy in data presentation (especially the "number" labels and percentage totals), would greatly improve the clarity of this information.

Line 173 refers to Table 4, which does not appear to be included in the document. Inclusion of this table would be essential to a full understanding of your findings.

Table 3 would benefit from more detailed commentary to explain its relevance and findings.

In the Discussion section, starting with a concise summary of the results obtained would provide a clear starting point for your analysis and interpretations.

The information between lines 210-218 should be supported by appropriately conducted tests of statistical significance. Consistency of terminology between text and tables is also important for reader understanding.

For lines 219-225, it would strengthen your argument to support your claims with specific results.

The interpretation of the results presented in lines 224-225 seems to require more robust analytical procedures to support claims of validity or reliability.

Finally, there appears to be an incorrect citation on line 229. Correcting this would help maintain the academic integrity of your paper.

Overall, your manuscript has the potential to make a significant contribution to the field. These suggestions are intended to improve the clarity, coherence, and rigor of your presentation, thereby increasing the impact of your research.

Comments on the Quality of English Language

Some sentences are unintelligible and should be rephrased. The text contains misquotes.

Author Response

Dear Reviewer,

Thank you for your suggestions. Following your guidance, we have made some changes to our article.

  • We added a more explicit articulation of the research questions and hypotheses between lines 92-96.
  • We provided additional information regarding the statistical tests used between lines 141-147.
  • We included more data in the results section, introducing the ß coefficient next to the p-value and citations to tables between lines 184-187 (formerly line 162).
  • Regarding anonymity, we first searched for the email address of the midwives' coordinator and then sent them the link to the questionnaire, which includes a dedicated section for informed consent. Personal data were separated from the study data and analyzed anonymously (lines 111-113/148-149).
  • We added more information regarding the ANOVA test result in Table 2 (lines 165-166, formerly lines 143-145).
  • We used linear regression, showing a statistically significant inverse correlation, specifically with the completion of studies in the Compassion Satisfaction subscale (β=-1.571, p=0.028) (formerly line 154, now 174).
  • The sentences reported between lines 163-165 were a transcription error that we deleted.
  •  We made a complete revision of all tables, as per your suggestion. We included the missing table, which is now Table 5 in this version (line 195, formerly 173).
  • We added more explanations regarding Table 3 between lines 178-182.
  •  We introduced a concise summary in the discussion section (lines 242-244).
  • Regarding the information reported between lines 267269 (formerly 210-218), we showed the data of the statistical test on Table 2.
  • We made some changes to the sentences between formerly lines 219-225, now lines 278-284.
  • Regarding the citation on formerly line 229, we based our sentences in this part of the study on Leinweber et al. (2017): "The majority of midwives (67%) recalled that their witnessed index traumatic birth event involved at least one care-related interpersonal event feature, and more than one-third recalled a birth event consisting of interpersonal care-related trauma features (disrespectful, poor, or abusive care) only (38%). An event consisting of at least one non-interpersonal feature (death or injury) was recalled by 61% of midwives, and 32% recalled an event consisting of non-interpersonal features exclusively. An event that involved both interpersonal and non-interpersonal trauma features was recalled by 30% of midwives."

Thank you again for your helpful comments.

Kind Regards,

Dr. Guzzon

Round 2

Reviewer 3 Report

Comments and Suggestions for Authors

Dear Authors,

After reviewing the revised manuscript, I am pleased to note that all of the previously raised comments and suggestions have been thoroughly and appropriately addressed. Your commitment to improving the paper and your attention to detail are commendable.

I have no further comments at this time. It has been a pleasure to follow the development of this manuscript. I wish you the best of luck in your future endeavors and am confident that your work will make a valuable contribution to the field.

Best regards,